# Chikungunya Outbreak in the Republic of the Congo, 2019—Epidemiological, Virological and Entomological Findings of a South-North Multidisciplinary Taskforce Investigation

**DOI:** 10.3390/v12091020

**Published:** 2020-09-13

**Authors:** Francesco Vairo, Martin Parfait Aimè Coussoud-Mavoungou, Francine Ntoumi, Concetta Castilletti, Lambert Kitembo, Najmul Haider, Fabrizio Carletti, Francesca Colavita, Cesare E. M. Gruber, Marco Iannetta, Francesco Messina, Simone Lanini, Biez Ulrich Judicaël, Emanuela Giombini, Chiara Montaldo, Chantal Portella, Steve Diafouka-Diatela, Martina Rueca, Richard Kock, Barbara Bartolini, Leonard Mboera, Vincent Munster, Robert Fischer, Stephanie Seifert, César Muñoz-Fontela, Beatriz Escudero-Pérez, Sergio Gomez-Medina, Emily V. Nelson, Patrick Kjia Tungu, Emanuele Nicastri, Vincenzo Puro, Antonino Di Caro, Maria Rosaria Capobianchi, Jacqueline Lydia Mikolo, Alimuddin Zumla, Giuseppe Ippolito

**Affiliations:** 1National Institute for Infectious Diseases ‘Lazzaro Spallanzani’, IRCCS, 00149 Rome, Italy; francesco.vairo@inmi.it (F.V.); concetta.castilletti@inmi.it (C.C.); fabrizio.carletti@inmi.it (F.C.); francesca.colavita@inmi.it (F.C.); cesare.gruber@inmi.it (C.E.M.G.); Marco.Iannetta@inmi.it (M.I.); francesco.messina@inmi.it (F.M.); simone.lanini@inmi.it (S.L.); emanuela.giombini@inmi.it (E.G.); chiara.montaldo@inmi.it (C.M.); martina.rueca@inmi.it (M.R.); barbara.bartolini@inmi.it (B.B.); emanuele.nicastri@inmi.it (E.N.); vincenzo.puro@inmi.it (V.P.); antonino.dicaro@inmi.it (A.D.C.); maria.capobianchi@inmi.it (M.R.C.); 2Fondation Congolaise Pour la Recherche Médicale (FCRM), Brazzaville CG-BZV, Congo; dl.congo@unesco-delegations.org; 3Ministry of Science and Technology, Brazzaville CG-BZV, Congo; 4University Marien Ngouabi, Brazzaville CG-BZV, Congo; 5Institute for Tropical Medicine, University of Tübingen, 72074 Tübingen, Germany; diafkietelas@fcrm-congo.com; 6Ministry of Public Health, Brazzaville CG-BZV, Congo; litchea@cg.afro.who.int (L.K.); sante@gouv.cd (B.U.J.); jlmikolo@gmail.com (J.L.M.); 7The Royal Veterinary College, University of London, Hawkshead Lane, Hertfordshire NW1 0TU, UK; nhaider@rvc.ac.uk (N.H.); rkock@rvc.ac.uk (R.K.); 8Department de Sante Publique, Pointe Noire CG-16, Congo; port22chant@gmail.com; 9SACIDS Foundation for One Health, Sokoine University of Agriculture, Morogoro 30007, Tanzania; leonard.mboera@sacids.org; 10Virus Ecology Unit, Laboratory of Virology, Rocky Mountain Laboratories, NIAID/NIH, Bethesda, MD 20814, USA; vincent.munster@nih.gov (V.M.); robert.fischer@nih.gov (R.F.); stephanie.seifert@nih.gov (S.S.); 11Bernhard Nocht Institute for Tropical Medicine, Bernhard Nocht Strasse, D-20359 Hamburg, Germany; munoz-fontela@bnitm.de (C.M.-F.); escudero.perez@bnitm.de (B.E.-P.); gomez-medina@bnitm.de (S.G.-M.); nelson-emily@bnitm.de (E.V.N.); 12German Center for Infection Research (DZIF), Partner Site Hamburg, 38124 Hamburg, Germany; 13National Institute for Medical Research, Dar es Salaam 9653, Tanzania; hq@nimr.or.tz; 14Laboratoire National de la Santè Publique, Brazzaville CG-BZV, Congo; 15Division of Infection and Immunity, Center for Clinical Microbiology, University College London, London WC1E 6BT, UK; a.zumla@ucl.ac.uk; 16National Institute of Health Research Biomedical Research Centre at UCL Hospitals, London W1T 7HA, UK

**Keywords:** chikungunya, Republic of Congo, outbreak, *Aedes spp*, mosquito, arbovirus, ONE-HEALTH

## Abstract

The Republic of Congo (RoC) declared a chikungunya (CHIK) outbreak on 9 February 2019. We conducted a ONE-Human-Animal HEALTH epidemiological, virological and entomological investigation. Methods: We collected national surveillance and epidemiological data. CHIK diagnosis was based on RT-PCR and CHIKV-specific antibodies. Full CHIKV genome sequences were obtained by Sanger and MinION approaches and Bayesian tree phylogenetic analysis was performed. Mosquito larvae and 215 adult mosquitoes were collected in different villages of Kouilou and Pointe-Noire districts and estimates of *Aedes (Ae.)* mosquitos’ CHIKV-infectious bites obtained. We found two new CHIKV sequences of the East/Central/South African (ECSA) lineage, clustering with the recent enzootic sub-clade 2, showing the A226V mutation. The RoC 2019 CHIKV strain has two novel mutations, E2-T126M and E2-H351N. Phylogenetic suggests a common origin from 2016 Angola strain, from which it diverged around 1989 (95% HPD 1985–1994). The infectious bite pattern was similar for 2017, 2018 and early 2019. One *Ae. albopictus* pool was RT-PCR positive. The 2019 RoC CHIKV strain seems to be recently introduced or be endemic in sylvatic cycle. Distinct from the contemporary Indian CHIKV isolates and in contrast to the original Central-African strains (transmitted by *Ae. aegypti*), it carries the A226V mutation, indicating an independent adaptive mutation in response to vector replacement (*Ae. albopictus* vs *Ae. aegypti*).

## 1. Introduction

Re-emerging zoonotic infectious diseases cause much human suffering worldwide, particularly in West and Central Africa where major arbovirosis and other zoonotic epidemics keep occurring. The recent Ebola virus disease (EVD) epidemics in West Africa (2013–2016) and in the Democratic Republic of Congo (DRC) (2018–2020) highlighted that, once an infectious disease takes hold locally, the region and the rest of the world is put at risk [1,2]. This stresses the need of an effective north-south cooperation and of a ONE-HEALTH approach including human, environmental and animal health sectors, to effectively address recurrent and emergent zoonotic threats.

Chikungunya (CHIK) is a disabling and debilitating zoonotic disease of humans caused by the chikungunya virus (CHIKV). CHIKV is an alphavirus showing four different genotypes originally based on geographical regions: the West African genotype (Senegal and Nigeria), the East/Central/South African (ECSA) genotype, the Asian genotype and the Indian Ocean Lineage (IOL) genotype [3,4,5,6,7]. CHIKV was first isolated in southern Tanzania in 1952 from the serum of a patient during an outbreak of an exanthematous febrile disease. *Aedes* (*Ae.*) *aegypti* was identified as the vector [8]. Since then, an unnoticed and retrospectively diagnosed large outbreak affected Cameroon in 2006 [9] and Gabon a year later, with a total of 20,000 suspected cases. *Ae. albopictus* was identified as the main vector [10]. CHIKV isolates from the 2007 Gabon outbreak belonged to the ECSA lineage with the E1-A226V mutation [10], which increases the competence of *Ae. albopictus* as disease vector [11,12]. In 2011, the same ECSA lineage caused an outbreak in the Republic of Congo (RoC) with 11,083 cases (91% from Brazzaville and 9% from the Pool Department) [13,14]. CHIKV has also spread beyond its original locations (Americas, Africa and Indian subcontinent) to the northern hemisphere including Europe [15]. In February 2019, RoC government declared an outbreak of CHIK in Kouilou, Pool, and Bouenza regions.

The aim of our investigation is a better understanding of the epidemiological dynamics of the 2019 CHIK outbreak in RoC, in order to support the local response, as requested to the recently established PANDORA-ID-NET consortium (https://www.pandora-id.net/), by The Congolese Foundation for Medical Research (FCRM) and by The National Public Health Laboratory (LNSP).

## 2. Materials and Methods

### 2.1. Study Sites

Health districts in the RoC, Kouilou, Pointe-Noire, Bouenza, Niari and Pool departments. The surveillance system was enhanced for active case finding and contacts’ tracing. Data on patients’ age, sex, place of living, time of symptoms’ onset and clinical symptoms were collected, and a notification form was distributed to the health facilities for regular central notification of suspected and confirmed cases.

### 2.2. Study Case Definitions

Suspect CHIK case: Patient presenting with acute onset of fever and joint pain;

Case Under investigation: Any suspected CHIK case with an epidemiological link with an area of ongoing circulation without laboratory confirmation;

Confirmed CHIK case: Patient with positive CHIKV RT-PCR or/and positive indirect immune fluorescence assay (IFA) or/and positive rapid test;

Non CHIK case: individual with negative CHIKV-RT-PCR and IFA; negative Rapid test and IFA.

### 2.3. Laboratory Diagnostics

CHIKV viral genome and standard CHIKV antibody testing were performed. Viral RNA was extracted from serum using QIAamp Viral RNA Mini Kit (QIAGEN^®^, Milan, Italy), for qualitative detection of CHIKV RNA with an “in-house” real-time RT-PCR. Specific CHIK IgG and IgM were detected using indirect immune fluorescence assay (IFA) (Anti-chikungunya Virus IIFT, Euroimmun AG, Luebeck, Germany). To overcome logistical shortcomings and to be able to operate in the field, a rapid test for specific IgG and IgM detection was used (STANDARD F chikungunya IgM/IgG FIA SDBIOSENSOR, Chungcheongbuk, Republic of Korea), with sensitivity 97% and specificity 99%.

### 2.4. Genomic Analyses

Two near complete CHIKV genome sequences were obtained from serum samples of two acutely infected women, one living in Tchali, Pointe Noire, collected on the 20 of March (n. 1, 63 years old) and the other living in Nkoungou, Pointe Noire, collected on the 5 of March (n. 2, age not available). Symptoms started 3–4 days before sampling.

The virus sequence from patient n. 1 was obtained by Sanger sequencing according to previously published procedures [16,17]. Specifically, a 11,434 nt long sequence was amplified in 24 overlapping RT-PCR amplicons (cycling condition: 50 °C × 30′; 94 °C × 15′; 94 °C × 40″–56 °C × 40″–72 °C × 1′ (×40); 72 °C × 10′) and Sanger sequenced (CHIK/PNRCNG/LNSP_INMI1-2019, GenBank accession number MK935343). Primers used for RT-PCR and sequencing were designed based on the reference sequence of Pakistan-07/2016 CHIKV isolate complete genome (GenBank accession number: MF774617.1) and are listed in Appendix A. The virus sequence from patient n. 2 was obtained by next generation sequencing (NGS), with the handheld third generation Nanopore sequencing technology MinION (Oxford Nanopore Technologies, Oxford, UK), using the approach described for field investigation on Ebola virus [2]. In particular, multiple sequence specific RT-PCR reactions were performed, (cycling condition: 45 °C × 20′; 94 °C × 5′; 94 °C × 20″–58 °C × 20″–68 °C × 2.5′(×40); 68 °C × 10′), generating five overlapping amplicons, assembled in a 11,787 nt long sequence (CHIK/PNRCNG/LNSP_INMI2-2019, GenBank accession number MK935344). MinION sequencing of pooled amplicons was performed according to manufacturer’s instructions using the Ligation Sequencing kit (Nanopore SQK-LSK109) on a FLOMIN106 flowcell. Primers used for RT-PCR and sequencing are listed in Appendix A.

### 2.5. MinION Sequencing Data Analysis 

The raw sequencing data were archived on a portable hard disk and analyzed offline. Correction and assembly were performed using Canu program [18]. The longest contig was compared to the non-redundant database using Blastn [19]. The nearest genome (LC259094) was used as reference to perform a consensus sequence through Burrows–Wheeler Aligner (BWA-MEM) [20]. The sequence obtained by BWA was aligned to the longest contig, and ambiguities were solved.

### 2.6. Phylogenetic Analysis

The phylogenetic trees were built using Maximum-Likelihood (ML) method and Bayesian Maximum Credibility tree to date back the coalescent events; besides the CHIK/PNRCNG/LNSP_INMI1-2019 and CHIK/PNRCNG/LNSP_INMI2-2019 sequences, both trees included 60 additional full-length genome retrieved from the NCBI database, representing the 3 major described CHIKV lineages: ECSA (*n* = 44, including the Indian Ocean sub-lineage, *n* = 26), Asia-Caribbean (*n* = 12), and West African (*n* = 4). Evolutionary distances for ML were computed using the General Time Reversible model (GTR) with Gamma distribution, and bootstraps were generated using 500 replicates, using the MEGA X software 10.1.

Bayesian phylogenetic tree was inferred using Bayesian Markov chain Monte Carlo (MCMC) approach available in BEAST v1.10.4 [21]. General time reversible model plus Gamma distributed rates among sites (GTR + G) was used as nucleotide substitution model and independent MCMC runs were carried out for strict model, along with constant population size coalescent priors. Chains were conducted for at least 100 × 106 generations with sampling every 10,000 steps and burn-in 10 × 106 generations. The convergence of the MCMC was assessed by calculating for each parameter the ESS (accepted if ESS > 250). Maximum clade credibility tree was obtained from the trees’ posterior distributions with the Tree-Annotator software v 1.10.4 [21].

### 2.7. Mutational Analysis

Mutational analysis for ORF 1 and ORF 2 proteins was performed using the prototype 1953 Tanzanian strain S27 (GenBank accession number: AF369024) as reference. Uniqueness of amino acid substitutions, first identified by comparison with the full genome sequences included in the phylogenetic tree, was verified by performing BLASTp search for 21 amino acid long peptides spanning the location of each substitution of interest, considering the first 100 hits from all (both full genome and partial) CHIKV sequences available in GenBank.

### 2.8. Estimation of Infectious Bites of CHIKV by Aedes Mosquito

We developed a biological process model and a mechanistic transmission model that follow the biological event of vectors and hosts independently as previously described for a different pathogen [22,23]. The model followed a cohort of mosquitos each day of a transmission season until they die and estimated the daily survival rate, the biting rate, and the Extrinsic Incubation Period with the mathematical equations provided earlier (Table 1). The model ran for one cohort of mosquitos at a time, starting with the cohort that bit on the first day of the selected period, and ran progressively through the remaining days of the season. The model then estimated the number of secondary infected hosts, also known as infectious bites (IB) from one infectious individual via the mosquito vectors. In absence of monitored mosquito data, the model assumed that there would be one mosquito each day of the year. Meteorological data (temperature, rainfall and humidity) from 1 January 2017 to 22 March 2019 were obtained from the Agence Nationale de l’Aviation Civile (ANAC) of Congo.

### 2.9. Statistical Analysis

Statistical analyses were performed using STATA 13.1 Descriptive statistics were used by means of proportions and median (interquartile range). Quantitative data were presented through frequency and percentage.

## 3. Results

### 3.1. Patients’ Characteristics

Demographic and clinical characteristics of the patients are reported in Table 2.

During the period between the declaration of the outbreak and 27 March (end of the investigation), 349 suspect patients were reported and samples sent to LNSP. Of these, 288 (82.5%) were residing in the Pointe Noire Department (Appendix A), 60 (17.2%) in the Kouilou Department and 1 in Dolisie (Niari Department). Of the 349 suspect patients, 84 (24·1%) were confirmed, 194 (55·6%) were under investigation and 71 (20.3%) were non-cases. Of the 278 confirmed and under investigation cases, 244 (87.8%) were from Pointe Noire, 33 (11.9%) from Kouilou and one from Dolisie. Of the 84 confirmed cases, 59 (70.3%) were positive at RT-PCR, 18 (21.4%) were IgM- and IgG-positive and 7 (8.3%) were IgM-only positive. The epidemic curve shows an increase of reported cases after strengthening of the surveillance system during the mission (Figure 1).

### 3.2. Molecular Characterization

ML phylogenetic analysis of the two new CHIKV sequences sharing 99.8% nucleotide identity has been later confirmed by Fritz et al. [24] reporting a strain belonging to the ECSA lineage, clustering with the more recent enzootic ECSA 2 sub-clade and showing A226V mutation. 

Figure 2 shows the Bayesian maximum clade credibility tree. Sequences harboring the A226V mutation are highlighted by black dots. The Bayesian tree analysis supports the assignment of both RoC 2019 CHIKV sequences to a unique strain belonging to ECSA 2 sub-clade. Hu/Angola/NIID54/2016 (GenBank accession number: LC259094), a strain isolated in 2016 in Angola [25] lacking the A226V mutation, is the closest related sequence (98.9% nucleotide identity) to both 2019 RoC sequences. The analysis suggests that these strains have a common origin, dating back to around 1973 (95% HPD 1970–1977), from which they diverged around 1989 (95% HPD 1985–1994).

SAn extensive analysis of the pattern of mutations carried by the 2019 RoC sequences, among those mutations reported to affect vectorial competence of CHIKV [5,12,26], is shown in Table 3.

The strains not carrying the A226V mutation show a high degree of variation, with mutations located at multiple sites along the ORF-1 and ORF-2 polyproteins. In contrast, the strains carrying the A226V mutation show less variability and share a common mutational pattern. In addition to the considered amino acid positions, the CHIK/PNRCNG/LNSP_INMI1-2019 and CHIK/PNRCNG/LNSP_INMI2-2019 strains carry additional unique mutations, in particular E2-T126M and E2-H351N, which are not present in any of the GenBank sequences archived so far.

### 3.3. Predictive Model

#### 3.3.1. Infectious Bites

We estimated IB for the period of 2017–2018 and in the first 2 months of 2019. The pattern was similar for the years 2017, 2018, and relevant 2019 period (Figure 3). If weather pattern remains similar to the previous two years, the current epidemic will continue until July, then the transmission will drop down for a couple of months and will increase again by early November. A plausible explanation for the absence of disease in 2018, despite a similar pattern of IB through 2017–2019 predicted by the model, is that the pathogen could have been absent until late 2018, when it could have been introduced by an infected migrant or spill over from a sylvatic host. This explanation is consistent with the phylogenetic analysis showing no evidence of evolving pattern from the strain previously circulating in the country (2011 RoC strain).

#### 3.3.2. Entomological Investigations

During the first week of March 2019, 215 adult mosquitoes were collected in different villages of Kouilou and Pointe-Noire districts. Further, on the third week of March, 38 adult mosquitoes were collected in Diosso (Kouilou department), of which 24 were *Ae. albopictus*, 2 *Ae. aegypti* and 12 other *Culicinae spp*. For PCR analysis purposes, the insects were pooled by collection site (Table 4).

A total of 100 houses were inspected for the presence of potential mosquitos breeding sites and for immature mosquitoes (larvae and pupae) outside the houses. Of these, 88 (88%) houses had water-holding containers in their surroundings including plastic containers, tires, water tanks and swamps. Water holding containers in 94 houses were found to harbor mosquito pupae. The overall containers, houses and Breteaux indices were 84%, 92% and 8.6%, respectively.

## 4. Discussion

Our study provides important insights on the epidemiological, virological and entomological characteristics of the 2019 CHIK outbreak in RoC, through a ONE HEALTH south–north multidisciplinary taskforce investigation. Patients’ demographic characteristics show a higher proportion of females probably due to their routine housekeeping activities, where most of the breeding sites are located in small containers for water collection. The median age is higher than in 2011 outbreak (34 vs 29 years) [13], and the proportion of cases increases with age, with children under 5 years old almost not affected. Age is a proxy-factor for specific behaviors causing higher exposure to *Ae. albopictus* bites (i.e., staying outdoors during daytime) or less tendency toward individual protection (i.e., use of repellents) in elderly people. Elderly people are also more likely to show symptoms and to access health structures than young people. The age profile could be also explained by the high susceptibility of the population with no herd immunity, supporting the hypothesis of absent, or very scarce circulation of the virus in the affected departments in the previous years. The median time between onset of symptoms and notification is much lower than in previous African and European outbreaks [15]. This could be related to a more severe initial symptomology, mainly joint pain, which brought patients to rapidly attend a health facility. The clinical presentation should be better clarified through dedicated cohort studies.

The phylogenetic analysis, based on near full genome sequences obtained from two patients living in Pointe Noire District, shows that the strain involved in this outbreak belongs to the ECSA enzootic lineage, specifically to a recent cluster, also called ECSA 2 [13]. This cluster includes CHIKV strains currently circulating in Central Africa and differentiates from the ECSA 1 also for the presence of the A226V mutation in some strains. Same findings were obtained from a patient in Diosso in the District of Kouilou [24]. The closest related is a strain isolated in 2016 in Angola, that lacks the A226V mutation, from which divergence seems to have occurred around 1990. In addition, RoC 2019 sequences belong to the same sub-clade of the 2011 RoC outbreak strain [13,14] but are placed on a distinct branch of the phylogenetic tree. A lower degree of variability was observed between strains carrying the A226V mutation and those that do not. In addition, two novel mutations, E2-T126M and E2-H351N, render the RoC 2019 strain unique. Adaptive changes due to other different mutations in E2 glycoprotein have been previously reported [27,28], and further studies are needed to explore the effect on viral infectivity and transmission possibly linked to the observed E2 mutations. The replacement of *Ae. aegypti* by *Ae. albopictus* already noticed previously (i.e., Gabon 2007 and Cameroon 2006) was also reported in RoC where the invasive species are predominant over the native ones in all locations except Brazzaville, suggesting that *Ae. albopictus* is displacing *Ae. aegypti* across RoC [29]. The 2019 RoC strain, like the strains from Gabon and Cameroon outbreaks (Chik_Cam_7079, GenBank_EF051584) [9,10], originates from the Central-African lineage, and it is distinct from the contemporary Indian/Indian Ocean isolates, but in contrast to original Central-African strains (transmitted by *Ae. aegypti*), it carries the A226V mutation. This implies an independent adaptive mutation in response to vector replacement (*Ae. albopictus vs Ae. aegypti*), confirming previous findings [5,11,26]. The virus might have been recently introduced into the area or be endemic in a sylvatic cycle, without spill over to humans in the years 2017–2018. It is important to evaluate the possibility of an endemic circulation of CHIKV at the human–animal interface. The presence of CHIKV should be investigated in wild (non-human primates, rodents, etc.), domestic animals (cows, goats) and wild-type vectors (*Ae. africanus*, *Ae. furcifer* and *Ae. taylori*). Human exposure should be assessed through longitudinal seroprevalence studies at the human–animal interface. The model highlights a suitable environment and shows potential for a CHIK outbreak in all studied years (2017, 2018 and first 2 months of the year 2019). The absence of outbreaks in previous years may indicate mere absence of the pathogen in the country or a low level of transmission into the vector not sufficient to spill over into the community. Our phylogenetic characterization is compatible with the introduction from other Central African countries. The virus could have been introduced from other endemic areas through the seaport city of Pointe Noire. Alternatively, the virus could have circulated among the wildlife for long before to reach the threshold of prevalence needed to cause human epidemics. Diosso, one of the first villages reporting CHIK cases in 2019, is an ecological hub for a potential spillover event, being surrounded on three sides by a large canyon, named Gorge of Diosso, visited by local and international visitors and inhabited by several animal species including pigs, bats, rats, mice, shrews, porcupine, snakes, squirrels and monkeys. We suggest further exploration of the role of wildlife in the CHIK sylvatic cycles. In Senegal, non-human primates were found to play an important role in maintenance of CHIKV in wildlife [30]. The predominance of *Ae. Albopictus* as dominant vector in the study area could be due to the specific ecological requirements of the two species. While *Ae. aegypti* prefers to occupy human houses (close and/or inside the human dwellings), *Ae. albopictus* prefers vegetation and forest areas more protected by control interventions. The entomological indexes where all very high, underlying the importance of community education about control of mosquito activity.

The limitations of the study include the scarce proactive countrywide human and entomological surveillance during the mission and the limited precision of the geolocalization, since data about patients’ residence were referred to a general zone (village or quartier). Strengthening the ongoing surveillance system is key for evaluating the spread of the disease, allocating the appropriate resources for outbreak response, evaluating clinical management of patients, advising on vector control measures and guiding preparedness for future outbreaks. Implementation of an active surveillance through community-based activities is a key issue. Reporting of suspect cases by community health workers, door-to-door case findings and subsequent diagnosis should be regularly implemented. The integration of arboviruses surveillance into the well-established and ongoing surveillance activities for malaria should be taken into consideration exploiting experience, tools and model from malaria surveillance. In addition, surveillance and research into the sylvatic reservoirs of known and unknown arboviruses is key in understanding the spatio-temporal dynamics of CHIKV and potentially newly emerging arboviruses, as is the case in the neighboring DRC where Dengue virus and CHIKV were found during investigation of a yellow fever outbreak [31].

## 5. Conclusions

The PANDORA model of south–north cooperation and the ONE HEALTh approach to tackle the CHIKV outbreak in RoC was successful allowing a fast, innovative and functional investigation and giving rapid and prompt insights into the outbreak’s dynamics. This paves the way for further collaborations among the involved stakeholders for short-, medium- and long-term epidemic response and preparedness plans.

## Figures and Tables

**Figure 1 viruses-12-01020-f001:**
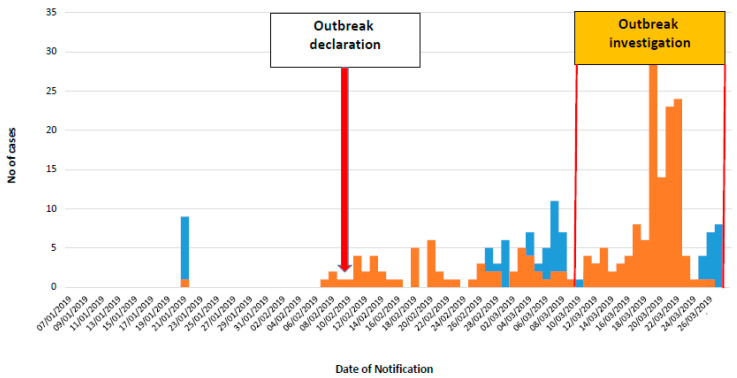
Epidemic curve of confirmed and under investigation cases. The epidemic curve shows an increase of reported cases after strengthening of the surveillance system during the mission (orange bars: cases under investigation, blue bars: confirmed cases).

**Figure 2 viruses-12-01020-f002:**
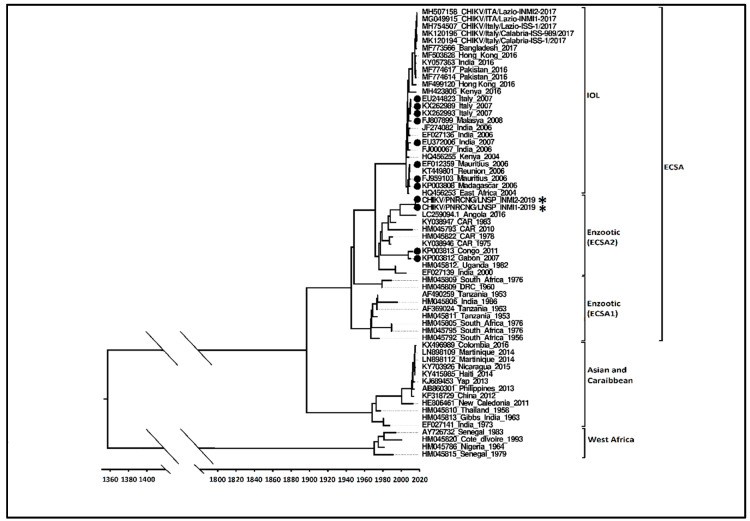
Bayesian maximum clade credibility tree built with full genome sequences of CHIK/PNRCNG/LNSP_INMI1-2019 and CHIK/PNRCNG/LNSP_INMI2-2019 strains highlighted by as an asterisk (*), in the context of 60 full genome sequences representing the 3 major CHIKV lineages: ECSA (*n* = 44, including the Indian Ocean sub-lineage, *n* = 26; the IOL sub-lineage includes the Pakistani-Italian 2017 cluster, *n* = 12 and the Italian 2007 cluster, *n* = 3), Asia-Caribbean (*n* = 12), and West Africa (*n* = 4). Each record consists of accession number, place and year of detection/isolation. The parameters used in BMCC were mutation model General Time Reversible + G, strict clock model and constant size demographic model. It was tested through MCMC for at least 100 × 10^6^ generations. The bar represents time coalescent in years. Sequences harboring the A226V mutation are highlighted by black dots.

**Figure 3 viruses-12-01020-f003:**
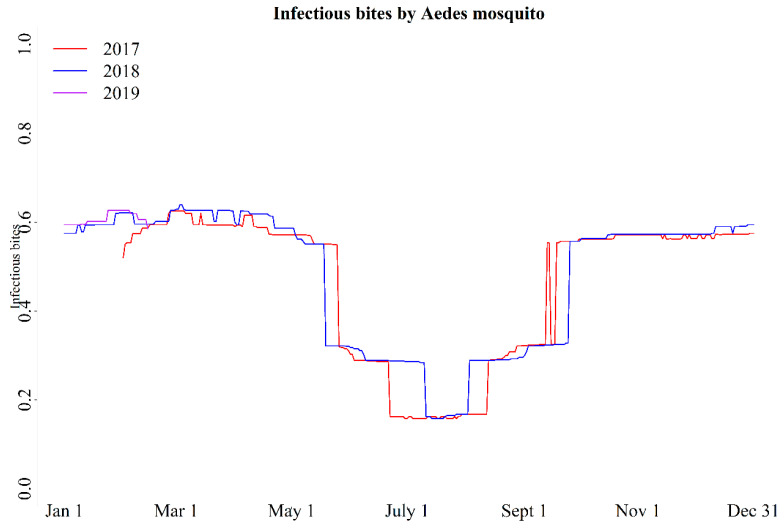
The Infectious bites of *Aedes* mosquito estimated from a mathematical model considering that mosquitoes were infected with chikungunya virus and exposed to environmental temperature, which was similar to the recorded meteorological temperature at a local weather station at Pointe-Noire, Republic of Congo.

**Table 1 viruses-12-01020-t001:** The biological parameters and the equations used in the model.

Traits	Source	Ref
Mosquitoes daily survival rate	e^−1/(−4.4 + 1.31*Tmean − 0.03*(Tmean)^2)^	[24]
Biting rate	(1/(0.0943 + 0.0043*T))/24	[25]
Extrinsic incubation period	(−0.1393 + 0.008T)/24	[5]
Viremic period	4 days	[26]
Mosquitoes to human transmission rate	0.8	[26]
Human to mosquito transmission rate	0.8	[26]
Probability of finding a susceptible human host by *Aedes spp* mosquito	0.5	[26]

Tmean = Daily mean temperature. T = Hourly temperature.

**Table 2 viruses-12-01020-t002:** Characteristics of patients (*N* = 349) by case classification.

Characteristics *	Total *n* (%)	Confirmed *n* (%)	Suspect *n* (%)	Non Case *n* (%)	*p* Value
**Department (*N* = 349)**					0.217
Pointe–Noire	288 (100)	58 (20.1)	186 (66.6)	44 (15.3)	
Kouilou	60 (100)	25 (41.7)	8 (13.3)	27 (45.0)	
Other	1 (100)	1 (100)	0	0	
**Gender (*N* = 302)**					0.217
Male	113 (100)	25 (22.1)	65 (57·5)	23 (20.4)	
Female	189 (100)	37 (19.6)	126 (66·7)	26 (13.8)	
**Age (*N* = 212)**					
Median (IQR)	34 (13–47)	41 (30–49)	34 (13–46)	33 (13–48)	
**Age (*N* = 212)**					0.7
≤5 years	4 (100)	0	3 (75.0)	1 (25.0)	
6–15 years	60 (100)	2 (3.3)	54 (90.0)	4 (6.7)	
16–40 years	68 (100)	4 (5.9)	59 (86.8)	5 (7.4)	
>40	80 (100)	6 (7.5)	70 (87.5)	4 (5.0)	
**Duration of symptoms (*N* = 248), Median (IQR)**	3 (2–4)	3 (2–4)	2 (1–4)	3 (2–5)	

* For each data set, the number of patients with available records is specified in brackets.

**Table 3 viruses-12-01020-t003:** Mutational profile of CHIK/PNRCNG/LNSP_INMI1-2019 and CHIK/PNRCNG/LNSP_INMI2-2019 (in bold characters) compared to the 60 full genome sequences included in the phylogenetic analysis (see Figure 2). The upper part of the table includes sequences carrying the original amino acid (A) at position E1-226; the lower part includes sequences with the E1-A226V mutation. The 1953 Tanzanian strain (S27), used as reference, is in the first row.

Polyprotein	ORF 1								ORF 2													
Amino acid position	476	665	680	1705	1857	1918	1948	2144	62	404	489	530	589	637	700	711	1020	1035	1078	1093	1126	1131
Protein	nsP1	nsP2		nsP3		nSP4			C	E2							E1					
Amino acid position	476	130	145	372	524	55	85	281	62	79	164	205	264	312	375	386	211	226	269	284	317	322
Strains non carrying the A226V mutation																						
AF369024 Tanzania 1953	**P**	**H**	**E**	**D**	**R**	**S**	**R**	**V**	**R**	**G**	**A**	**G**	**V**	**T**	**S**	**V**	**K**	**A**	**M**	**D**	**I**	**V**
LC259094 Angola 2016										E	T								V			A
MF503628 Hong Kong 2016		Y	D			N				E	T		A	M	T	A	E		V	E		A
EF027136 India 2006										E	T			M	T	A			V	E		A
JF274082 India 2006					R					E	T			M	T	A			V	E		A
FJ000067 India 2006										E	T			M	T	A			V	E		A
EF027139 India 2000					R					E	T								V			A
HM045793 CAR??										E	T								V			A
AF490259 Tanzania 1953					R					E												.
HM045813 Gibbs India 1963	.									E		D					E					A
KY703926 Nicaragua 2015										E		D					E					A
KJ689453 Yap 2013										E		D					E					A
KY415985 Haiti 2014										E		D					E					A
LN898109 Martinique 2014										E		D					E					A
LN898112 Martinique 2014										E		D					E					A
HM045786 Nigeria 1964							G			E									V			A
AY726732 Senegal 1983							G			E									V			A
HM045809 South Africa 1976	Q									E									V			A
HM045795 South Africa 1976										E												A
KY057363 India 2016		Y	D			N				E	T		A	M	T	A	E	.	V	E	V	A
HM045811 Tanzania 1953										E												A
HM045821 Senegal 1963					R																	.
HM045810 Thailand 1958										E		D					E		.			A
HM045815 Senegal 1979							G			E		.							V			A
HM045820 Cote dIvoire 1993							G			E		.							V			A
AB860301 Philippines 2013										E		D					E					A
KX496989 Colombia 2016										E		D					E					A
EF027141 India 1973										E		D					E					A
HE806461 New Caledonia 2011										E		D					E					A
KF318729 China 2012										E		D					E					A
HQ456255 Kenya 2004										E	T			M	T	A			V	E		A
MG049915 CHIKV/ITA/Lazio-INMI1-2017		Y	D	E		N			C	E	T	S	A	M	T	A	E		V	E	V	A
MH507158 CHIKV/ITA/Lazio-INMI2-2017		Y	D	E		N			C	E	T	S	A	M	T	A	E		V	E	V	A
MF774617 Pakistan-07/2016	Q	Y	D			N		I		E	T		A	M	T	A	E		V	E	V	A
MF774614 Pakistan-04/2016		Y	D			N		I		E	T		A	M	T	A	E		V	E	V	A
MF773566 Bangladesh 2017		Y	D	E		N		.		E	T	S	A	M	T	A	E		V	E	V	A
MF499120 Hong Kong 2016		Y								E	T		A	M	T	A	E		V	E	V	A
HM045805 South Africa 1976										E			X									A
HM045792 South Africa 1956										E												A
MH423806 Kenya 2016										E	T		A	M	T	A	E		V	E		A
HQ456253 East Africa 2004										E	T			M	T	A			V	E		A
HM045809 DRC 1960	Q									E									V			A
HM045822 CAR 1978										E	T								V			A
KY038946 CAR 1975										E	T								V			A
KY038947 CAR 1983	Q									E	T								V			A
HM045812. Uganda 1982										E	T								V			A
MK120194 CHIKV/Italy/Calabria-ISS-1/2017		Y	D	E		N			C	E	T	S	A	M	T	A	E		V	E	V	A
MK120196 CHIKV/Italy/Calabria-ISS-989/2017		Y	D	E		N			C	E	T	S	A	M	T	A	E		V	E	V	A
MH754507 CHIKV/Italy/Lazio-ISS-1/2017		Y	D	E		N			C	E	T	S	A	M	T	A	E		V	E	V	A
Strains carryng the A226V mutation																						
**CHIK/PNRCNG/LNSP_INMI2-2019**										**E**	**T**	**D**						**V**	**V**			**A**
**CHIK/PNRCNG/LNSP_INMI1-2019**										**E**	**T**	**D**						**V**	**V**			**A**
KT449801 Reunion 2006										E	T			M	T	A		V	V	E		A
EU244823 Italy 2007										E	T			M	T	A		V	V	E		A
KX262993 Italy 2007										E	T			M	T	A		V	V	E		A
KX262989 Italy 2007					R					E	T			M	T	A		V	V	E		A
KP003813 Congo 2011										E	T							V	V			A
KP003812 Gabon 2007										E	T							V	V			A
FJ807899 Malasya 2008										E	T			M	T	A		V	V	E		A
EU372006 India 2007										E	T			M	T	A		V	V	E		A
FJ959103 Mauritius 2006										E	T			M	T	A		V	V	E		A
EF012359 Mauritius 2006										E	T			M	T	A		V	V	E		A
KP003808 Madagascar 2006	Q				R					E	T			M	T	A		V	V	E		A

**Table 4 viruses-12-01020-t004:** Mosquitoes collected by collection period, location, species and RT-PCR results.

Collection Time	Department	Village/Quartier	Identified Species (*n*)	RT-PCR
1st week of March	Kouilou	Nkoungou	*Ae. albopictus* (18)	Positive
		Diosso	*Ae. albopictus* (37)	Negative
		Mengo	*Ae. albopictus* (8)	Negative
		Matombi	*Ae. albopictus* (16)	Negative
		Mabindou	*Ae. albopictus* (25)	Negative
1st week of March	Pointe Noire	Mpita	*Ae. albopictus* (34)	Negative
			*Ae. aegypti* (20)	Negative
		Tchiamba Nzassi	*Ae. albopictus* (47)	Negative
		Siafoumou	*Ae. albopictus* (10)	Negative
3rd week of March	Kouilou	Diosso	*Ae. aegypti* (2)	Negative
			*Ae. albopictus* (24)	Negative
			*Culicinae spp* (12)	Negative

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
