# Peer review of "Chikungunya Outbreak in the Republic of the Congo, 2019—Epidemiological, Virological and Entomological Findings of a South-North Multidisciplinary Taskforce Investigation"

_viruses, 2020, doi:10.3390/v12091020_

Round 1

Reviewer 1 Report

In the article by numerous authors including the Pandora-ID-NET taskforce, the authors present

data accumulated from the 2019 Chikungunya virus (CHIKV) outbreak in the democratic republic of the Congo.

Interestingly, new CHIKV sequences in the ECSA lineage were identified.  It will be of interest to explore what

adaptive changes if any, the observed E2 mutations confer.  Taken together the manuscript provides useful information

to the CHIKV community and highlights the importance of having these kinds of taskforces in place.  A major critique

that should be addressed in order for the article to be published is the low quality of the figures in the manuscript.  For

example,  Figure 1 should have more distinction between confirmed and under investigation to better aid the reader.  The

inclusion of Figure 2 is unclear and appears to be a supplemental material at best.  Figures 3 and 4 are very low quality,

the text is barely legible.  These issues must be addressed for further consideration.  

Author Response

Response to Reviewer 1

Comments of reviewer 1:

In the article by numerous authors including the Pandora-ID-NET taskforce, the authors present data accumulated from the 2019 Chikungunya virus (CHIKV) outbreak in the democratic republic of the Congo.

Interestingly, new CHIKV sequences in the ECSA lineage were identified. 

  1. It will be of interest to explore what adaptive changes if any, the observed E2 mutations confer. 

Taken together the manuscript provides useful information to the CHIKV community and highlights the importance of having these kinds of taskforces in place. 

A major critique that should be addressed in order for the article to be published is the low quality of the figures in the manuscript.  

  1. Figure 1 should have more distinction between confirmed and under investigation to better aid the reader. 
  2. The inclusion of Figure 2 is unclear and appears to be a supplemental material at best. 
  3. Figures 3 and 4 are very low quality, the text is barely legible. 

Response to reviewer 1:

  1. We thank the Reviewer for the suggestion. So far, we could not investigate possible adaptive changes linked to the new E2 mutations as the viral isolate was not available. To our knowledge, other different E2 mutations are reported in literature as responsible of adaptive changes in infectivity for Aedes mosquitoes (doi: 1371/journal.pone.0006835; https://doi.org/10.1016/j.virol.2016.06.025). Therefore, it would be interesting to perform an in-silico analysis in a further study and a more specific evaluation will be necessary to elucidate the role of the E2 mutations observed in this strain. We included a statement in the discussion (line 468-470) to underline this aspect and we added references 28 and 29.  
  2. Figure 1 has been revised accordingly with the suggestion of the reviewer.
  3. Figure 2 has been eliminated since it was not useful for the comprehension of the manuscript.
  4. Figures 3 and 4 (current figures 2 and 3) have been replaced with a better quality ones.

Reviewer 2 Report

The 2019 Chikungunya Outbreak in the Republic of the Congo – Epidemiological, Virological, and Entomological findings of a ONE HEALTH south north multidisciplinary taskforce investigation

In the manuscript, Vairo and colleagues provided important insights into the RoC 2019 CHIK outbreak through epidemiology, virology and entomology. At the same time, the author used more methodological details and phylogenetic analysis methods to verify the source of the virus strain and find the mutant strain. However, some content was still concerned, and hope that the following comments can be revised.

Methods

The study design follows the standard of the field, is logical, and is appropriately designed to achieve their goals.

  1. Line 116 and 265---In the recent clinical research data shown that the sickness rates were highest in the age group 25–64 years. However, the age segmentation (table 1) is not divided in this way. Is there any reference?
  2. Line 133---In laboratory diagnosis, the rapid test is mentioned as an indicator tool for confirming whether there is Chikungunya disease. Can you further mark the specificity and sensitivity of this rapid test in the Methods?

Results

  1. At present, the figures are very low resolution, insufficient for professional publication. We understand this might be a reflection of the reviewer portal, however in either case, higher quality (resolution) images should be provided prior to publication.
  2. The colors used to denote suspect and confirmed in figure 2 are not clear, at the same time, the color of the cluster size is similar, that they are easily misidentified. We recommend fixing this.
  3. Line 213---Figure S1 does not appear in the article. Please confirm if there are any omissions when uploading the file.
  4. Line 320---Whether I have misunderstood, Figure 4 illustrates IB in the period of 2017-2018 and in the first 2 months of 2019, without describing the temperature-related tests. However, the legend seems to be inconsistent with the content of this article and the figure, please confirm it.
  5. Line 326---I can't find the description of Figure 5 in the main text. Please remove it if you don't need it. If you want to use it, please explain in this article.

Discussion/Conclusions

  1. CHIK transmitted via bites from infective female Aedes mosquitoes and can rapidly spread in the past few decades. This has been driven in part by climate change and increased rates of global trade and international travel.

It is recommended that the author not only use the epidemiological, virological and entomological to analyze the 2019 Congo outbreak, maybe further explore the role of travelers in the Congo outbreak.

  1. Line 412 to 414, I cannot understand what the author wants to express, can explain clearly or delete it.

General Comments

  1. Line 24--- Correspondence: orresponding authors: → corresponding
  2. Line 105--- norther → northern
  3. Line 265---Please align each item and data in Table 1, and modify it in accordance with the submission guidelines.
  4. Line 305---What is the definition of NB?
  5. Line 454---Reference documents should be revised in accordance with the submission guidelines. In addition, please reconfirm some references. For example: reference 6,7,18,30,32,33.
  6. For the legend in Figure 1 and Figure 2, can describe more and may let readers understand more clearly the meaning that the author wants to express.

Author Response

Comments of reviewer 2:

In the manuscript, Vairo and colleagues provided important insights into the RoC 2019 CHIK outbreak through epidemiology, virology and entomology. At the same time, the author used more methodological details and phylogenetic analysis methods to verify the source of the virus strain and find the mutant strain. However, some content was still concerned, and hope that the following comments can be revised.

 Methods

The study design follows the standard of the field, is logical, and is appropriately designed to achieve their goals.

  1. Line 116 and 265---In the recent clinical research data shown that the sickness rates were highest in the age group 25–64 years. However, the age segmentation (table 1) is not divided in this way. Is there any reference?
  2. Line 133---In laboratory diagnosis, the rapid test is mentioned as an indicator tool for confirming whether there is Chikungunya disease. Can you further mark the specificity and sensitivity of this rapid test in the Methods?

Responses to reviewer 2:

  1. We have chosen these age groups knowing from the literature that Chikungunya fever affects all age groups and we wanted to see if there was a difference within the large 25-64 range.
  2. As requested, we added the information on specificity and sensitivity of the rapid test at line 202 (sensitivity 97%, specificity 99%).

Comments of reviewer 2:

Results

  1. At present, the figures are very low resolution, insufficient for professional publication. We understand this might be a reflection of the reviewer portal, however in either case, higher quality (resolution) images should be provided prior to publication.
  2. The colors used to denote suspect and confirmed in figure 2 are not clear, at the same time, the color of the cluster size is similar, that they are easily misidentified. We recommend fixing this.
  3. Line 213---Figure S1 does not appear in the article. Please confirm if there are any omissions when uploading the file.
  4. Line 320---Whether I have misunderstood, Figure 4 illustrates IB in the period of 2017-2018 and in the first 2 months of 2019, without describing the temperature-related tests. However, the legend seems to be inconsistent with the content of this article and the figure, please confirm it.
  5. Line 326---I can't find the description of Figure 5 in the main text. Please remove it if you don't need it. If you want to use it, please explain in this article.

Responses to reviewer 2:

  1. Figure 1, 3 and 4 have been replaced with a better quality ones and revised accordingly with the suggestion of another reviewer. Figures 3 and 4 became the current figures 2 and 3.
  2. Figure 2 has been eliminated since it was not useful for the comprehension of the manuscript, as suggested by another reviewer.
  3. Figure S1 has been renamed table 4 and the reference to the table is in the text at line 269.
  4. Yes, Fig 4 (current figure 3) illustrates estimated the infectious bites of Aedes mosquitoes in the period of 2017-2018 and first two month of 2019. We have corrected the figure legends that now reads

“The Infectious bites of Aedes mosquito estimated from a mathematical model considering that mosquitoes were infected with chikungunya virus and exposed to environmental temperature which was similar to the recorded meteorological temperature at a local weather station at Pointe-Noire, Republic of Congo.”

  1. Figure 5 has been removed.

Comments of reviewer 2:

Discussion/Conclusions

  1. CHIK transmitted via bites from infective female Aedes mosquitoes and can rapidly spread in the past few decades. This has been driven in part by climate change and increased rates of global trade and international travel. It is recommended that the author not only use the epidemiological, virological and entomological to analyze the 2019 Congo outbreak, maybe further explore the role of travelers in the Congo outbreak.
  1. Line 412 to 414, I cannot understand what the author wants to express, can explain clearly or delete it.

Responses to reviewer 2:

  1. It is a good idea for a next investigation but unfortunately we don’t have this information regarding the 2019 outbreak.
  2. We have deleted lines 412, 413 414.

Comments of reviewer 2:

General Comments

  1. Line 24--- Correspondence: orresponding authors: → corresponding
  2. Line 105--- norther → northern
  3. Line 265---Please align each item and data in Table 1, and modify it in accordance with the submission guidelines.
  4. Line 305---What is the definition of NB?
  5. Line 454---Reference documents should be revised in accordance with the submission guidelines. In addition, please reconfirm some references. For example: reference 6,7,18,30,32,33.
  6. For the legend in Figure 1 and Figure 2, can describe more and may let readers understand more clearly the meaning that the author wants to express.

Responses to reviewer 2:

  1. Line 24 (current 23): we have corrected orresponding authors with corresponding
  2. Line 105 (current line 136): we have corrected norther with northern
  3. Line 265 (current 360): Items and data have been aligned data in Table 1, and the table has been modified in accordance with the submission guidelines.
  4. NB is the written abbreviation used before a pieceof important information to make certain that readers notice It is Latin for "note well" and comes from the Latin roots notāre ("to note") and bene ("well") (https://dictionary.cambridge.org/dictionary/english/nb). Anyway the sentence has been removed together with figure 2.
  5. Reference have been revised and corrected. References 8, 9, 20, 27, 28, 29 have been removed.
  6. The legend of figure 1 has been expanded to better explain the meaning of the figure itself. Figure 2 has been removed.

Reviewer 3 Report

The manuscript “The 2019 Chikungunya Outbreak in the Republic of the Congo – Epidemiological, Virological, and Entomological findings of a ONE HEALTH south-north multidisciplinary taskforce investigation” provides insight into the recent CHIK RoC outbreak, including the evolutionary origin of this virus strain. The CHIKV genome sequences obtained from this work provide unique additions to virus databases. I have some minor suggestions for manuscript improvement before publication:

Title:

Should be shortened.

Abstract:

Please write out the full phrase: East/Central/South African (ECSA).

The methodology pertaining to the mosquito species surveyed should be elaborated on: One Ae. albopictus pool was positive out of how many? What was the extent of the species and areas surveyed for mosquito species? This information should be included in the abstract.

Introduction:

The second sentence is a run on, and should be split into two. The first sentence of the second paragraph also has this same issue.

A final paragraph should be added, introducing the specific study aims. The last two sentences of the current last paragraph should be moved into this new third paragraph.

Materials and Methods:

Primers used for RT-PCR, and cycling conditions should be added. If it is the same as the cited references, please make sure this is stated. Primers used to generate the amplicons should also be given.

IFA is defined after the abbreviation is already used, please change.

Results:

For the paragraph starting on page 5, line 216, the methods associated with the Bayesian analysis can be removed, as it is ready mentioned in the methods.

Table 1: Can the placement of characteristics descriptions be better centered in the table?

Table 2: I would try to reduce the amount of grey highlighting. I think that it isn’t necessary to have a grey highlight for the reference sequence, maybe make the residues bold instead to distinguish it.

Table 3: Should remove column 4 and in the column “Identified species” just add “n=” in parenthesis to list the numbers.

Figure 1: Need to make the font size larger and bolded black, it is a bit hard to see dates and legend.

Figure 2: What are the green dots? Are the dot colors just the boarder of the dots? The dot coloring should be made clearer. ­­­

Figure 3: Instead of using the arrow to show the strains of the present study, perhaps make these sequences bold?  Or use an asterisk next to each. Make sure this is designated in the legend.

Figure 4 and 5: Please change the green color. It is almost impossible to see.

Where is Fig. S1?

Author Response

Comments of reviewer 3

The manuscript “The 2019 Chikungunya Outbreak in the Republic of the Congo – Epidemiological, Virological, and Entomological findings of a ONE HEALTH south-north multidisciplinary taskforce investigation” provides insight into the recent CHIK RoC outbreak, including the evolutionary origin of this virus strain. The CHIKV genome sequences obtained from this work provide unique additions to virus databases. I have some minor suggestions for manuscript improvement before publication:

  1. Title: Should be shortened.

Abstract:

  1. Please write out the full phrase: East/Central/South African (ECSA).
  2. The methodology pertaining to the mosquito species surveyed should be elaborated on: One  albopictus pool was positive out of how many? What was the extent of the species and areas surveyed for mosquito species? This information should be included in the abstract.

Response to reviewer 3:

  1. The title has been shortened and changed with: “Chikungunya Outbreak in the Republic of the Congo, 2019– Epidemiological, virological and entomological findings of a south-north multidisciplinary taskforce investigation”.
  2. ECSA have been written as East/Central/South African
  3. The available information have been added in the abstract (line 90-91).

Comments of reviewer 3

Introduction:

  1. The second sentence is a run on, and should be split into two. The first sentence of the second paragraph also has this same issue.
  2. A final paragraph should be added, introducing the specific study aims. The last two sentences of the current last paragraph should be moved into this new third paragraph.

Materials and Methods:

  1. Primers used for RT-PCR, and cycling conditions should be added. If it is the same as the cited references, please make sure this is stated. Primers used to generate the amplicons should also be given.
  2. IFA is defined after the abbreviation is already used, please change.

Response to reviewer 3:

  1. These sentences have been split in 2.
  2. A final paragraph has been added, introducing the specific study aim.
  3. As requested, primers and cycling conditions have been included (see lines 210-214, 218, 225-26)) and supplementary annexes 1 and 2 about the primers used to generate the amplicons have been added
  4. IFA definition has been provided at line 192, the first time is mentioned.

Comments of reviewer 3

Results:

  1. For the paragraph starting on page 5, line 216, the methods associated with the Bayesian analysis can be removed, as it is ready mentioned in the methods.
  2. Table 1: Can the placement of characteristics descriptions be better centered in the table?
  3. Table 2: I would try to reduce the amount of grey highlighting. I think that it isn’t necessary to have a grey highlight for the reference sequence, maybe make the residues bold instead to distinguish it.
  4. Table 3: Should remove column 4 and in the column “Identified species” just add “n=” in parenthesis to list the numbers.
  5. Figure 1: Need to make the font size larger and bolded black, it is a bit hard to see dates and legend.
  6. Figure 2: What are the green dots? Are the dot colors just the boarder of the dots? The dot coloring should be made clearer. ­­­
  7. Figure 3: Instead of using the arrow to show the strains of the present study, perhaps make these sequences bold?  Or use an asterisk next to each. Make sure this is designated in the legend.
  8. Figure 4 and 5: Please change the green color. It is almost impossible to see.
  9. Where is Fig. S1?

Response to reviewer 3:

  1. The methods associated with the Bayesian analysis have been removed
  2. Placement of characteristics descriptions has better been centered in the table 1
  3. Table 2 has been changes removing the grey highlighting. CHIK/PNRCNG/LNSP_INMI1-2019 and CHIK/PNRCNG/LNSP_INMI2-2019 are now written in bold characters.
  4. Table 3 has been modified according the reviewer suggestions.
  5. Figure 1 has been modified according the reviewer suggestions.
  6. Figure 2 has been deleted as suggested by another reviewer since it was not useful for the understanding of the manuscript.
  7. Figure 3 (current figure 2) has been modified as suggested by the reviewer, deleting the arrow and making the new sequences bold and marked with an asterisk next to each. The legend has been modified accordingly.
  8. Figure 4 (current figure 3) has been replaced with a better quality one and figure 5 has been removed.
  9. Figure S1 has been renamed table 4 and the reference to the table is in the text at line 269

Round 2

Reviewer 1 Report

All concerns have been adequately addressed.

Author Response

  • English language and style have been reviewed and spell errors have been corrected (as track changes in the manuscript)
  • Fig. 1 and 2 have been provided at better resolution
  • Misspellings in figure 1 have been corrected 
  • All references to mosquito species are italicized throughout the manuscript